# The Evolution of Therapies Targeting Bruton Tyrosine Kinase for the Treatment of Chronic Lymphocytic Leukaemia: Future Perspectives

**DOI:** 10.3390/cancers15092596

**Published:** 2023-05-03

**Authors:** Toby A. Eyre, John C. Riches

**Affiliations:** 1Oxford Cancer and Haematology Centre, Oxford University Hospitals NHS Foundation Trust, Churchill Hospital, Headington, Oxford OX3 7LE, UK; 2Centre for Haemato-Oncology, Barts Cancer Institute, Queen Mary University of London, London EC1M 6BQ, UK

**Keywords:** chronic lymphocytic leukaemia, Bruton tyrosine kinase, ibrutinib, acalabrutinib, zanubrutinib, pirtobrutinib, nemtabrutinib, NX-2127, NX-5948

## Abstract

**Simple Summary:**

The development of drugs that successfully target Bruton tyrosine kinase (BTK) represents a major scientific and clinical advance for the treatment of chronic lymphocytic leukaemia (CLL). This article will begin by reviewing the scientific observations that underpinned the targeting of this kinase in CLL. It will then discuss the evolution of BTK inhibitors, from the initial studies with ibrutinib, to the development of more specific and noncovalently binding BTK inhibitors, focusing on how different agents can be sequenced in patients who are resistant or intolerant to one of these drugs. Finally, this article will also review the concept of BTK degraders and offer insights into the future direction of the field.

**Abstract:**

The development of inhibitors of Bruton tyrosine kinase (BTK) and B-cell lymphoma 2 (BCL2) has resulted in a paradigm shift in the treatment of chronic lymphocytic leukaemia (CLL) over the last decade. Observations regarding the importance of B-cell receptor signalling for the survival and proliferation of CLL cells led to the development of the first-in-class BTK inhibitor (BTKi), ibrutinib, for the treatment of CLL. Despite being better tolerated than chemoimmunotherapy, ibrutinib does have side effects, some of which are due to the off-target inhibition of kinases other than BTK. As a result, more specific inhibitors of BTK were developed, such as acalabrutinib and zanubrutinib, which have demonstrated equivalent/enhanced efficacy and improved tolerability in large randomized clinical trials. Despite the increased specificity for BTK, side effects and treatment resistance remain therapeutic challenges. As these drugs all bind covalently to BTK, an alternative approach was to develop noncovalent inhibitors of BTK, including pirtobrutinib and nemtabrutinib. The alternative mechanisms of BTK-binding of these agents has the potential to overcome resistance mutations, something that has been borne out in early clinical trial data. A further step in the clinical development of BTK inhibition has been the introduction of BTK degraders, which remove BTK by ubiquitination and proteasomal degradation, in marked contrast to BTK inhibition. This article will review the evolution of BTK inhibition for CLL and offer future perspectives on the sequencing of an increasing number of different agents, and how this may be impacted on by mutations in BTK itself and other kinases.

## 1. Introduction

The development of inhibitors of B-cell receptor (BCR) signalling, particularly those that inhibit Bruton tyrosine kinase (BTK), has transformed the treatment landscape for chronic lymphocytic leukaemia (CLL) over the last decade. The recognition of the importance of this signalling pathway for CLL was due to a number of important observations over the preceding 10–15 years. One of the most important findings was that the mutational status of the variable region of the immunoglobulin heavy chain (*IGHV*) gene of the CLL BCRs was of prognostic value, with patients with unmutated (>98% homology to germline) *IGHV* genes (UM-*IGHV*) having a poorer prognosis than those with mutated (≤98% homology to germline) *IGHV* genes (M-*IGHV*) [1]. This prognostic difference is reflected in differences in the behaviour of CLL cells in vitro. UM-*IGHV* CLL cells have higher expression of surface immunoglobulin (Ig; usually IgM), which is associated with a retained ability to undergo calcium flux and tyrosine phosphorylation upon BCR ligation [2,3]. A further important observation was that CLL cells exhibit nonrandom “stereo-typed” BCRs, consistent with selection pressure by antigen. In theory, the biological complexity of normal humoral immune responses should allow for the production of a huge range (>1 × 10^9^) of BCRs, meaning that the chances of two patients with CLL having identical or near-identical BCRs should be negligible [4]. Despite this, up to a third of CLL cases have BCRs with very similar sequences—much too high for this to be a chance occurrence [4,5,6]. This suggests that antigenic stimulation is an important component of the pathogenesis of CLL, driving selection of nonrandom BCRs that recognize particular epitopes on these antigens. Notably, BCRs from M-*IGHV* CLL cases are thought to have higher-affinity binding to restricted sets of epitopes, reflecting the fact that the clone has arisen from a B-cell that has undergone somatic hypermutation and affinity maturation in the germinal centre [7]. In contrast, BCRs from UM-*IGHV* CLL cases express polyreactive BCRs that bind with low affinity to environmental and autoantigens such as vimentin, myosin, or rheumatoid factors [7,8,9]. Intriguingly, further reports provide evidence that CLL BCRs themselves can mimic engagement by antigen [10]. One mechanism by which the binding of extracellular antigen to the BCR is communicated to the intracellular signalling apparatus is by inducing clustering of BCRs, facilitating the formation of “microsignalosomes” [11]. The observation that CLL BCRs are able to bind to each other and cluster in the absence of antigen, enabling them to signal in an autonomous manner, is further evidence of the importance of this pathway. Even more support for this has been provided by gene expression profiling studies highlighting BCR signalling as the most differentially upregulated pathway in CLL cells activated in lymph nodes [12].

The BCR complex is composed of a membrane immunoglobulin (IgM) which is bound noncovalently to a CD79a/CD79b heterodimer [13]. When the IgM is engaged by antigen (or by another BCR in the case of CLL), the SRC family kinases LYN and spleen tyrosine kinase (SYK) act to phosphorylate tyrosine resides in the cytoplasmic immune-receptor tyrosine-based activation motif domains of CD79a/CD79b [14,15]. This allows recruitment of the other components of the signalosome, which includes BTK, the guanine exchange factor, VAV, and the adaptor proteins growth factor receptor-bound protein 2 and B-cell linker (BLNK) in addition to LYN and SYK [16]. SYK phosphorylation of BLNK allows recruitment of phospholipase C gamma 2 (PLC-γ2), which is phosphorylated by BTK and SYK to produce the second messenger diacylglycerol (DAG) and inositol-triphosphate (IP3) [17]. DAG activates protein kinase C which is responsible for many of the downstream effects of BCR signalling, while IP3 mediates calcium flux [16]. A further important component of BCR signalling is the activation of phosphoinositide 3-kinase (PI3K), which phosphorylates phosphatidylinositol 4,5-bisphosphate to create phosphatidylinositol 3,4,5-triphosphate, enabling the recruitment of BTK and other kinases to continue BCR activation.

The molecular understanding of BCR signalling highlights multiple potential targets. Several drugs that inhibit kinases involved in this pathway entered clinical testing including inhibitors of SYK (fostamatinib, entospletinib), pan-SRC kinases (dasatinib), PI3K (idelalisib, IPI-145, ACP319), and BTK (ibrutinib, spebrutinib (CC-292), and tirabrutinib (ONO-4059)) [7,13,18]. Despite all of these agents showing efficacy to a greater or lesser degree, BTK and PI3K rapidly became the preferred targets, with ibrutinib and idelalisib as the initial lead compounds, respectively. One of the notable clinical features observed with all BCR signalling inhibitors, from the first trials of the SYK inhibitor fostamatinib onwards, was a rapid reduction in lymphadenopathy/splenomegaly accompanied by a lymphocytosis [19]. This phenomenon represents a redistribution of the CLL cells from the secondary lymphoid organs to the peripheral blood, resulting in an efflux of tumour cells from the protective nodal microenvironment, which, along with a direct proapoptotic effect, underlies the clinical efficacy of these agents [20]. This review will focus on BTK-targeting therapies and how they have evolved over the last 10–15 years (Figure 1).

## 2. Ibrutinib: The “First in Class” BTKi

Ibrutinib, initially known as PCI-32765, was the first BTKi to enter clinical testing. Ibrutinib is an orally bioavailable small molecule which binds irreversibly to a cysteine residue (C481) in the BTK kinase domain to inhibit its enzymatic activity [21]. BTK could already be inferred as a critical component of BCR signalling and therefore humoral immunity due the observations made by Bruton regarding X-linked agammaglobulinemia (XLA) [22,23,24]. Individuals with XLA have mutations in the *BTK* gene (present on the long arm of the X-chromosome) and an immunodeficiency characterized by the absence of mature B-cells, resulting in severe antibody deficiency and recurrent infections that can manifest as soon as the protective effect of maternal immunoglobulins wanes at around 3–6 months of age. Preclinical observations confirmed the importance of BTK in CLL, demonstrating overexpression of this kinase in the leukemic cells compared with healthy B-cells, with ibrutinib demonstrating preferential, albeit modest, killing of CLL cells in vitro [25].

An initial phase 1b/2 study in 85 patients with relapsed/refractory CLL or small lymphocytic lymphoma (SLL) compared two different doses of ibrutinib, 420 mg versus 840 mg [26]. Interestingly, the overall response rate was identical in the two groups, at 71%, with continuous therapy generally well tolerated. Post-treatment pharmacokinetic assessments indicated full occupancy of BTK by ibrutinib at both dose levels, establishing 420 mg once daily as the standard dose in CLL/SLL. Importantly, the response rate of patients with deletions of the short arm of chromosome 17 (del17p) was comparable at 68%, highlighting the effectiveness of ibrutinib in patients who had historically had a poor prognosis with chemoimmunotherapy [27]. These initial findings were supported by the three RESONATE studies. The first RESONATE study demonstrated a significant improvement in both progression-free survival and overall survival (OS) with the use of ibrutinib when compared with the anti-CD20 antibody ofatumumab in patients with relapsed or refractory (R/R) CLL or SLL [28,29]. This was subsequently investigated in the frontline setting in the RESONATE-2 study, which also demonstrated a significant in improvement in PFS and OS with ibrutinib when compared with chlorambucil [30]. With a median follow-up of 60 months, PFS was 70% for ibrutinib versus 12% for chlorambucil, with ibrutinib also improving 5-year OS [31]. Ibrutinib was also well tolerated, with 42% of patients continuing to remain on ibrutinib at 8 years in a recently reported follow-up analysis [32]. Finally, the RESONATE-17 trial confirmed the efficacy of ibrutinib in patients with del17p, with 75% of patients remaining alive at 2 years—a significant improvement compared to historical controls [27,33]. While these initial trials could be criticized for comparing ibrutinib against relatively ineffective monotherapies, the superiority of ibrutinib has been confirmed by further studies versus chemoimmunotherapy. For example, the Eastern Cooperative Oncology Group–American College of Radiology Imaging Network (ECOG–ACRIN) study demonstrated an improvement in PFS and OS with ibrutinib–rituximab when compared with fludarabine, cyclophosphamide, and rituximab (FCR) [34,35]. Notably, a subgroup analysis suggested that the survival benefit is greater in patients with UM-*IGHV* genes, a finding replicated in other studies, logically suggesting that BTK inhibition is more effective in cases where the CLL cells have a higher BCR signalling capacity [2,3,36].

Initial studies also demonstrated the efficacy of the PI3K inhibitor, idelalisib, for the treatment of CLL [37]. A randomized phase 3 study compared idelalisib–rituximab to rituximab monotherapy in R/R CLL, showing an improvement in PFS and OS in the idelalisib arm [38]. While this led to the approval of idelalisib for CLL, it became apparent with further follow-up that this agent increases the incidence of several immune-related and infectious side effects including diarrhoea and colitis, hepatitis, CMV reactivation, and pneumonitis [39]. The higher incidence of adverse events was also observed with another PI3K inhibitor, duvelisib, suggesting a class effect and leading to black box warnings for both agents [40]. However, the efficacy of the idelalisib–rituximab combination does mean that it remains an option for some CLL patients, particularly those who are unsuitable for, or who are refractory to, BTKi- and venetoclax-based treatment [41,42]. Despite the fact that ibrutinib demonstrates a better safety profile than chemoimmunotherapy such as FCR, it is not without side effects. Common adverse events include diarrhoea, cough, infection, and myalgia, with bleeding another frequent and sometimes severe problem. One of the particular areas of concern with this agent is its cardiovascular effects: hypertension, atrial fibrillation, ventricular arrhythmia, and sudden cardiac death [43,44]. A critical feature is that while ibrutinib is an effective inhibitor of BTK, it also inhibits a wide variety of other kinases such as interleukin-2-inducible T-cell kinase and epidermal growth factor receptor tyrosine kinase [45]. Notably, it is ibrutinib’s inhibition of another kinase, C-terminal SRC kinase, that is thought to be responsible for the increased risk of atrial fibrillation seen with this drug [46]. As patients will need to take a BTKi continuously long-term to maintain control of their disease, there was a strong rationale for the development of highly specific BTKis for the treatment of CLL.

## 3. Increasing Specificity for BTK: Acalabrutinib and Zanubrutinib

The advent of the second-generation BTKis acalabrutinib and zanubrutinib now provides additional options in the management of front-line and relapsed CLL. Acalabrutinib (dosed at 100 mg twice daily) is a highly selective irreversible covalent BTKi licensed as monotherapy in R/R CLL following the ASCEND trial which demonstrated its superiority over investigator choice of either idelalisib–rituximab or bendamustine–rituximab [47,48]. Acalabrutinib is also now approved with/without the anti-CD20 monoclonal antibody obinutuzumab in the front-line setting following the ELEVATE-TN trial [49]. Both acalabrutinib monotherapy and acalabrutinib–obinutuzumab showed a substantial, superior 5-year PFS advantage compared with chlorambucil–obinutuzumab [50]. The addition of obinutuzumab in this study provides a 12% improvement in 5-year PFS compared to acalabrutinib monotherapy. Notably, the benefit was seen in *TP53*-intact patients only, particularly in those with UM-*IGHV* CLL, replicating the earlier observations with ibrutinib [51]. Patients with mutations in *TP53* or del17p also obtained durable disease control with acalabrutinib-based treatment, with a 4-year PFS of 76% and 75%, respectively, for acalabrutinib monotherapy and acalabrutinib–obinutuzumab. Acalabrutinib-treated patients in the ASCEND trial obtained an equivalent PFS whether *TP53* mutated/del17p or not.

The ELEVATE-RR open-labelled randomized controlled trial directly compared acalabrutinib with ibrutinib in R/R CLL patients and at least one high-risk genetic aberrancy (del17p/*TP53* mutation or 11q deletion) [52]. The study was designed to assess the noninferiority of PFS and also a hierarchical superiority assessment of toxicity differences between the two BTKis. At a median follow up of 40.9 months, there was no difference in PFS between the two agents but acalabrutinib-treated patients experienced a broad improvement in safety profile. All-grade cardiac adverse events and noncardiac adverse events (diarrhoea, myalgia/arthralgia, bleeding) were all improved with acalabrutinib with fewer adverse events leading to discontinuation. In light of this, the improved specificity of acalabrutinib for BTK does appear to translate into an improved safety profile when compared to ibrutinib, while retaining comparable efficacy. Notably, acalabrutinib has also demonstrated efficacy and safety in ibrutinib-intolerant CLL patients and is a valuable treatment option in this specific patient group [53,54].

Zanubrutinib (dosed at 160 mg twice daily) is another second-generation irreversible covalent BTKi approved as monotherapy for both front-line and relapsed CLL following results of the SEQUOIA trial and the ALPINE trial, respectively [55,56]. Zanubrutinib was developed to ensure better BTK specificity than ibrutinib and more sustained BTK occupancy, with exposure coverage above the half-maximal inhibitory concentration (IC50) across a 24 h dosing period [57]. The SEQUOIA trial demonstrated an improved PFS for continuous zanubrutinib compared to fixed-duration bendamustine–rituximab in patients in the front-line setting without del17p [56]. At a median follow up of 26.2 months, the median PFS was not reached in either group, but the 24-month PFS was 69.5% and 85.5%, respectively. The benefit was once again primarily seen in patients with UM-*IGHV* CLL. One hundred and ten patients with del17p/*TP53* mutation were enrolled in a separate open label phase II nonrandomized arm of the SEQUOIA trial (Arm C), receiving zanubrutinib monotherapy to progression. At a median follow-up of 30.5 months, the overall response rate (ORR) was 90%, the estimated 24-month PFS was 88.9%, and the estimated 24-month overall survival was 93.6%.

The recently published ALPINE trial directly compared zanubrutinib monotherapy with ibrutinib monotherapy in a large open-label, phase 3, randomized trial with ORR (excluding partial response with lymphocytosis) the primary end point, with PFS and safety key secondary endpoints [55]. The population was broader than in ELEVATE-TN, with all BTKi-naïve relapsed CLL patients enrolled, and was a relatively low-risk patient cohort (median number of prior lines 1, TP53 mutated/17p deletion 23%). With a median follow-up of 29.6 months, zanubrutinib demonstrated a superior PFS compared to ibrutinib. This difference was also noted in a subgroup of patients with *TP53* mutated/del17p CLL (HR 0.53). Discontinuation rates for reasons other than progression, cardiac sudden deaths, and atrial fibrillation rates were all important findings in favour of zanubrutinib. The toxicity profile between the two agents for other parameters such as hypertension, infection, bleeding, and cytopenia rates were similar. Zanubrutinib also recently demonstrated safety and efficacy in 67 patients previously intolerant to either ibrutinib or acalabrutinib [58].

As a result, it now seems reasonable to recommend these second-generation molecules with increased specificity for BTK over ibrutinib, due to the improved safety profile with acalabrutinib and the demonstrated improved efficacy and safety with zanubrutinib. Whether zanubrutinib is superior in terms of efficacy compared to acalabrutinib is unknown, and there are no head-to-head clinical trials planned or enrolling. The ALPINE and ELEVATE-RR trials studied different patient risk populations, with different geographies, across different treatment eras, and it is therefore impossible to cross-compare these studies. However, now that the clinical benefits of targeting BTK with improved specificity are proven, the next step in the evolutionary process of inhibiting this kinase in CLL is to test whether altering the mode of binding of drugs to BTK can further enhance the “fitness” of these agents for treating patients.

## 4. Noncovalent BTK Inhibition: Pirtobrutinib and Nemtabrutinib

While covalent BTKis have dramatically improved outcomes for patients with CLL/SLL, they are not curative. Long-term continuous usage can be associated with side effects that are difficult for patients to tolerate and also result in resistance due to development of mutations. Several resistance mutations have now been described, including a cysteine-to-serine mutation in BTK at the binding site of ibrutinib (C481S mutation) or gain-of-function mutations in the downstream kinase PLC-γ2 (R665W and L845F mutations) [59,60]. As a result, noncovalent BTKis have been developed with the potential to overcome the most common resistance mechanism associated with covalent BTKi use. Noncovalent binding to BTK does not rely on the C481 binding site, and so drugs can be designed that inhibit wildtype and C481-mutant BTK with equivalent potency. The two agents that are most advanced in clinical development are pirtobrutinib and nemtabrutinib.

Pirtobrutinib (formally LOXO-305) is a highly selective, first-in-class noncovalent (reversible) BTKi evaluated for both safety and efficacy in patients with CLL/SLL previously treated with a covalent BTKi in the first-in-human phase 1–2 BRUIN trial [61]. Pirtobrutinib has a favourable pharmacokinetic profile with high oral bioavailability and an extended half-life (approximately 19 h). This enables once-daily dosing with sustained plasma drug levels throughout the 24 h dosing interval, regardless of the intrinsic rate of BTK turnover. The selectivity profile of pirtobrutinib has the potential to minimize off-target inhibition and associated toxicity. The most recent update of the CLL patients in the BRUIN trial provided survival and toxicity data on 247 BTKi pretreated CLL/SLL patients with a median of 19.4 months follow-up [62]. The ORR for all patients was 82.2% when including partial response with lymphocytosis. Similar response rates were seen in dual-exposed (covalent BTKi and BCL2i) patients, *TP53* aberrant patients, pentad-exposed patients (anti-CD20, chemotherapy, covalent BTKi, BCL2i, Pi3Ki), and C481-mutated patients. Perhaps unsurprisingly, response rates in patients with downstream mutations in PLCγ2 were lower (ORR 55.6%). The median PFS for the whole cohort was 19.6 months. The most common all-grade treatment-emergent adverse events (TEAEs) across all 773 B-cell malignancy patients in BRUIN treated with pirtobrutinib were fatigue (29%), neutropenia (24%), and diarrhoea (24%). The rates of BTKi-associated events of special interest were low, including hypertension (9.2%), atrial fibrillation/flutter (2.8%), minor bleeding, and major haemorrhage (2.2%) [63]. Notably, only 3% of patients across the whole of the BRUIN trial discontinued pirtobrutinib due to a treatment-related side effect. Pirtobrutinib also demonstrated an excellent safety profile in 123 patients with B-cell malignancies who stopped a prior covalent BTKi (ibrutinib (n = 118, 96%), acalabrutinib (n = 29, 24%), or zanubrutinib (n = 6, 5%)) due to intolerance. Overall, 7% of these 123 patients subsequently discontinued pirtobrutinib for adverse events (only four stopped for reasons related to pirtobrutinib), and recurrences of adverse events were generally at a lesser grade [64]. A number of ongoing randomized trials are enrolling to consolidate its role in covalent BTKi-exposed patients and potentially move the agent further up the treatment pathway, such as the BRUIN-322 study, which is comparing venetoclax–rituximab against the combination of venetoclax–rituximab–pirtobrutinib in BTK naïve and exposed patients [65]. Figure 2 sets out a possible future treatment algorithm for patients receiving BTKis in an era where pirtobrutinib is widely accessible for routine clinical use.

Nemtabrutinib (MK-1026, formerly ARQ-531) is another noncovalent BTK in earlier development in the phase I-II 1/2 BELLWAVE-001 study. Recently updated efficacy data for 57 CLL/SLL pts with CLL/SLL treated with nemtabrutinib 65 mg once daily and safety for all 112 patients with B-cell malignancies treated at the 65 mg dose were presented [66]. The median number of prior lines of treatment was 4 (1–18), with 95% of those enrolled having received a prior covalent BTKi, with 42% also having received a BCL2 inhibitor. At a relatively short median follow-up of 8.1 months, the ORR was 56% and estimated median duration of response was 24.4 months. Seventy-three percent of patients experienced any-grade treatment-related adverse events. The most common (≥10%) were dysgeusia (21%), neutropenia (20%), fatigue (13%), nausea and thrombocytopenia (12% each), and diarrhoea and hypertension (10% each). Treatment-related discontinuations occurred in 15 pts (13%), somewhat higher than seen in pirtobrutinib-treated patients in the BRUIN trial (2.6%). Nemtabrutinib is less selective than pirtobrutinib, and further follow-up and a larger sample size is required to understand the future role of this promising agent. The pivotal clinical trials of BTKis for the treatment of CLL are summarized in Table 1.

## 5. BTK-Degradation: NX-2127 and NX-5948

Another BTK-targeting approach that may prove to be a critical addition to the CLL armamentarium is the strategy of degrading this kinase. In contrast to the agents discussed above which inhibit BTK function, “BTK degraders” essentially remove the BTK protein from the cell by targeting it for degradation by the proteasome. Ubiquitin-dependent proteolysis is a major pathway that degrades intracellular proteins as part of normal cellular maintenance processes [67]. In this pathway, proteins are targeted for degradation by the proteasome by the transfer of ubiquitin molecules to the protein of interest (in this case, BTK) by the combined action of ubiquitin-activating enzymes, ubiquitin-conjugating enzymes, and E3 ubiquitin–protein ligases. The first approaches utilized an existing drug (e.g., the angiogenesis inhibitor ovalicin) attached to short phosphopeptides that could be recognized by one of these E3 ubiquitin–protein ligases [68]. A major advance was the discovery that another E3 ubiquitin ligase, cereblon, was the target of thalidomide and its analogies lenalidomide and pomalidomide, widely used for the treatment of multiple myeloma [69,70]. These agents modulate cereblon to target IKAROS Family Zinc Finger 1 (IKZF1) and IKZF3 for degradation resulting in their immunomodulatory and anticancer activity. Several BTK degraders are being developed, including NX-2127 and NX-5948, which have now entered early phase clinical trials. Notably, NX-2127 targets both BTK and IKZF3 while NX-5948 just selectively degrades BTK. Both are currently in phase 1 studies with NX-2127 already demonstrating clinical responses in heavily pretreated (median 6 prior therapies) patients with CLL, including those with BTK mutations resistant to both covalent and noncovalent BTKis [71]. The ability of these drugs to potentially overcome the resistance mutations that emerge with BTK *inhibitor* treatment may mean that they form an important component of future treatment algorithms for CLL [72].

## 6. Conclusions

The last decade has seen great advances in the treatment of CLL, with multiple covalent and noncovalent BTKis demonstrating significant clinical efficacy. However, there remain several questions and challenges. A key question is whether the activity of noncovalent BTKis will be equivalent in patients resistant to different covalent BTKis. The C481S mutations are most commonly seen with ibrutinib and acalabrutinib, with relatively little being known about zanubrutinib. However, recently, the novel Leu528Trp mutation has been described in a small case series of seven zanubrutinib-treated patients [73]. Two of these seven patients demonstrated clinical cross-resistance and progressive enrichment of the BTK Leu528Trp mutation over time with subsequent treatment with pirtobrutinib. If these data are replicated in larger cohorts, it may have implications for the role of BTK mutation testing in the future as well as the choice of covalent BTKis, particularly if pirtobrutinib receives regulatory approval in R/R CLL patients previously treated with a covalent BTKi. Clinical trials are ongoing to assess pirtobrutinib versus both ibrutinib (NCT05254743) and immunochemotherapy (bendamustine–rituximab) (NCT05023980) in the front-line setting. It will be important to establish whether the resistance mechanisms (on-target kinase domain BTK mutations (V416L, A428D, M437R, T474I, and L528W) and downstream PLCγ2 mutations) described with pirtobrutinib in patients exposed to covalent BTKis exposed patients will also occur in pirtobrutinib-treated patients who have not received a covalent BTKi [74]. It will also be critical to understand if these mechanisms lead to cross-resistance to covalent BTKis if used subsequently. If not, it may be possible to “reverse” the sequential order of BTK inhibition in the future—a further key unanswered question.

Finally, both covalent and noncovalent BTKis are being studied in combination in numerous ongoing clinical trials. The most common partner agents are anti-CD20 monoclonal antibodies (most commonly obinutuzumab) and BCL2 inhibitors. Treatment strategies in the front-line and relapsed settings have generally been to deliver treatment according to MRD-stopping rules or as fixed duration. The aim for both strategies is to stop therapy in a deep remission after a time-limited duration and therefore limit both toxicity and the induction of resistance mechanisms to both BTK and BCL2 inhibitors. Ibrutinib plus venetoclax as fixed duration in the front-line setting is closest to being broadly approved following the results of the CAPTIVATE and GLOW trials [75,76,77]. It remains an incompletely answered question as to what influence fixed-duration BTK inhibition will have on subsequent C481 mutation rates, the benefit derived from retreatment with a covalent BTKi, alternative resistance mechanisms, and time-to-resistance in patients treated with this approach. Despite these remaining questions, two decades of multitudinous preclinical and clinical observations have established BTK as a critical target for the treatment of CLL. Further refinement of novel agents in the context of well-designed clinical trials should provide the “selection pressure” for the continued evolution of BTK-targeting therapies for this disease.

## Figures and Tables

**Figure 1 cancers-15-02596-f001:**
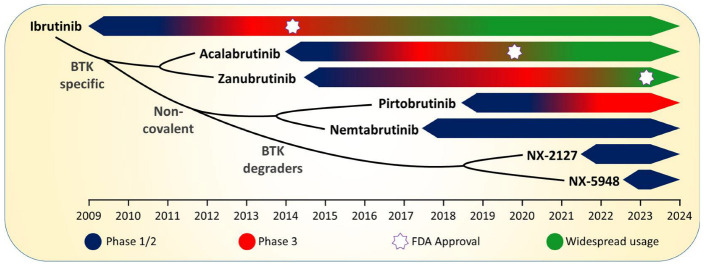
Timeline showing the evolution of BTK-targeting therapies for CLL. The timeline for phase 1/2 clinical testing is shown in dark blue, phase 3 clinical testing in red, with widespread adoption in green. A white star indicates the timing of Food and Drug Administration (FDA) approval.

**Figure 2 cancers-15-02596-f002:**
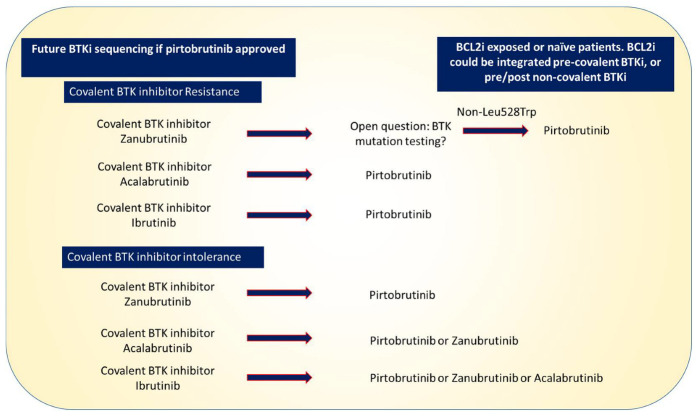
Potential future sequencing of BTKis if pirtobrutinib is approved. Pirtobrutinib represents a good option for patients developing resistance to any of the covalent BTKis—the role of mutation testing to guide therapy remains an open question. There are several options for patients who are intolerant of BTKis. Patients who are intolerant to ibrutinib could switch to a more specific covalent inhibitor or pirtobrutinib; pirtobrutinib is also a potential option for patients intolerant of acalabrutinib or zanubrutinib.

**Table 1 cancers-15-02596-t001:** Summary of pivotal clinical trials of BTKis for CLL.

BTK Inhibitor	Clinical Trial	ORR	PFS	OS	Discont. Rates/AEs	Ref.
Ibrutinib	RESONATE	91% (I)	Median PFS: 44.1 m (I) vs. 8.1 m (Ofa)	Median OS: not reached. Ofa group crossed over to I	Grade ≥ 3 AF: 6% (I) Grade ≥ 3 HTN: 8% (I)	[28,29]
	RESONATE-2	92% (I) vs. 37% (C)	5 y PFS: 70% (I) vs. 12% (C)	5 y OS: 83% (I) vs. 68% (C)	AF: 16% (I) HTN: 26% (I)	[30,31,32]
	ECOG-ACRIN	95.8% (IR) vs. 81.1% (FCR)	5 y PFS: 78% (IR) vs. 51% (FCR)	5 y OS: 95% (IR) vs. 89% (FCR)	Grade ≥ 3 AF: 4.5% (IR) vs. 0% (FCR) Grade ≥ 3 HTN: 11.4% (IR) vs. 1.9% (FCR)	[34,35]
Acalabrutinib	ASCEND	83% (A) vs. 84% (Idelalisib–R/BR)	42 m PFS: 62% (A) vs. 19% (Idelalisib–R/BR)	42 m OS: 78% (A) vs. 65% (Idela-R/BR)	AF: 8% (A) vs. 3% (Idela-R/BR) HTN: 8% (A) vs. 5% (Idela-R/BR) Discont: 23% (A) vs. 67% (Idela-R) vs. 17% (BR)	[47,48]
	ELEVATE-TN	96% (A-Obi) vs. 90% (A) vs. 83% (C-Obi)	5 y PFS: 84% (A-Obi) vs. 72% (A) vs. 21% (C-Obi)	5 y OS: 90% (A-Obi) vs. 84% (A) vs. 82% (C-Obi)	AF: 6.2% (A-Obi) vs. 7.3% (A) vs. 0.6% (C-Obi) HTN: 9.6% (A-Obi) vs. 8.9% (A) vs. 3.6% (C-Obi)	[49,50]
	ELEVATE-RR	81% (A) vs. 77% (I)	Median PFS: 38.4 m (A) vs. 38.4 m (I)	Median OS: not reached in either treatment group	AF: 9.4% (A) vs. 16% (I) HTN: 9.4% (A) vs. 23.2% (I) Discont: 14.7% (A) vs. 21.3% (I)	[52]
Zanubrutinib	SEQUOIA	94.6% (Z) vs. 85.3% (BR)	24 m PFS: 85.5% (Z) vs. 69.5% (BR)	24 m OS: 94.3% (Z) vs. 94.6% (BR)	AF: 3% (Z) vs. 3% (BE) Discont: 8% (Z) vs. 14% (BR)	[56]
	ALPINE	83.5% (Z) vs. 74.2% (I)	24 m PFS: 78.4% (Z) vs. 65.9% (I)	Median OS: not reached in either treatment group	C/D: 0 pts (Z) vs. 6 pts (I) AF: 5.2% (Z) vs. 13.3% (I) Discont: 62 pts (Z) vs. 92 pts (I)	[55]
Pirtobrutinib	BRUIN	74% (P)	Median PFS: 19.4 m	NE	Grade ≥ 3 AF: 1% (P) Grade ≥ 3 HTN: 3% (P) Discont: 2% (P)	[61,62]
Nemtabrutinib	BELLWAVE	56% (O)	Median PFS: 24.4 m	NE	HTN: 10% (N) Discont: 13% (N)	[66]

Abbreviations used in the table: I = ibrutinib; A = acalabrutinib; Z = zanubrutinib; P = pirtobrutinib; N = nemtabrutinib; Idela = idelalisib; C = chlorambucil; Ofa = ofatumumab; Obi = obinutuzumab; BR = bendamustine–rituximab; FCR = fludarabine–cyclophosphamide–rituximab; AF = atrial fibrillation; HTN = hypertension; C/D = cardiac death; discont. = discontinuation rate; pts = patients; m = months; y = years; NE = not evaluable.

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
