# Peer review of "The Evolution of Therapies Targeting Bruton Tyrosine Kinase for the Treatment of Chronic Lymphocytic Leukaemia: Future Perspectives"

_cancers, 2023, doi:10.3390/cancers15092596_

Round 1
Reviewer 1 Report
An excellent review
A few suggestions:
1. The BCR-induced signalling cascade described on page 2 should be accompanied by a Figure.
2. Line 113: be……..be?
3. Line 305: follow up?
Author Response
We thank the Reviewer for their complimentary comments regarding the manuscript. We did consider adding a figure showing the BCR-signaling cascade and activity of the various kinase inhibitors/degraders but decided against including this for two reasons. Firstly, multiple previous review articles have included such a figure and we do not feel that re-producing this would add anything to this particular review. Secondly, the review is focused on the one kinase, BTK, so including the full cascade is as a figure is slightly out-of-scope.
We thank the Reviewer for picking up the two errors and have amended these in the revised version.
Reviewer 2 Report
|
The authors describe the mechanism of action of BTK inhibitors, the activity and toxicity profile of covalent and non-covalent inhibitors, and BTK degraders. Critical issues related to the emergence of mutations that impair the activity of BTK inhibitors are reported and discussed, and future perspectives are commented on. |
|
|
|
|
|
1 |
The biological aspects related to the mechanism of action, off-target effects, and resistance are those of greater interest and clearly reported. To make the paper easier to read. clinical trials should be commented on with little detail on efficacy data. |
|
2 |
The quality of the figures should be improved. In particular, figure 2 is not clear. |
|
3 |
Line 202: (atrial fibrillation (9.4% vs. 16%)), hypertension (9.4% vs 23.2%)) Please, delete the double parenthesis and change vs with ‘vs.’ |
|
4 |
Lines 212, 215, 221: Please, change SEQUIOA with SEQUOIA. |
|
5 |
A review on BTK inhibitors should include a specific section focused on combinations with venetoclax. Combination regimens are only commented on in the discussion.
|
Author Response
We thank the Reviewer for reviewing the manuscript and for their comments. In response:
1) We feel that inclusion of the efficacy data is an important component of this manuscript as a comparison of the response and survival rates with these drugs is critical to their development. We agree that the details could be reduced to improve readability, so have removed much of the data previously in parentheses, instead incorporating this into a table.
2) We have amended the legend for figure 2 to improve the clarity of the figure to the reader.
3) We have removed the data in parentheses and incorporated this into the table as suggested by Reviewer 3.
4) We have corrected SEQUIOA to SEQUOIA.
5) We agree with the Reviewer that combinations of BTK-targeting therapies with other drugs such as venetoclax or anti-CD20 antibodies are of major clinical interest. However, the aim of this article was to review the evolution of therapies targeting BTK for CLL. As a result, we feel that a review of combination therapies (including those with venetoclax) is outside the scope of this manuscript.
Reviewer 3 Report
This review is very well written and conceptualized and covers all aspects of BTKi treatment currently available in the CLL/SLL literature. It guides through the topic very well and is written in a very precise manner.
However, although not necessary, I would like to see a graphical overview of BCR signaling in CLL as described in the introduction, including the various BTK-targeting agents.
Also, this review could benefit from a tabular overview comparing the different compounds, listing the respective studies with references, ORR, PFS, OS, discontinuation and adverse events.
Author Response
We thank the Reviewer for their complimentary comments regarding the manuscript. As per our response to Reviewer 1 we did consider adding a graphical overview of BCR-signaling in CLL including agents targeting this cascade, but decided against including this for the reasons previously outlined.
We thank the Reviewer for their suggestion regarding a tabular overview and have incorporated a table in the revised version.